# 4D-Former: Multimodal 4D Panoptic Segmentation

**Ali Athar**[1,3†∗]    **Enxu Li**[1,2∗]    **Sergio Casas**[1,2]    **Raquel Urtasun**[1,2]
[1]Waabi    [2]University of Toronto    [3]RWTH Aachen University
{aathar, tli, sergio, urtasun}@waabi.ai

**Abstract:** 4D panoptic segmentation is a challenging but practically useful task that requires every point in a LiDAR point-cloud sequence to be assigned a semantic class label, and individual objects to be segmented and tracked over time. Existing approaches utilize only LiDAR inputs which convey limited information in regions with point sparsity. This problem can, however, be mitigated by utilizing RGB camera images which offer appearance-based information that can reinforce the geometry-based LiDAR features. Motivated by this, we propose 4D-Former: a novel method for 4D panoptic segmentation which leverages both LiDAR and image modalities, and predicts semantic masks as well as temporally consistent object masks for the input point-cloud sequence. We encode semantic classes and objects using a set of concise queries which absorb feature information from both data modalities. Additionally, we propose a learned mechanism to associate object tracks over time which reasons over both appearance and spatial location. We apply 4D-Former to the nuScenes and SemanticKITTI datasets where it achieves state-of-the-art results. For more information, visit the project website: https://waabi.ai/4dformer.

**Keywords:** Panoptic Segmentation, Sensor Fusion, Temporal Reasoning, Autonomous Driving

## 1 Introduction

Perception systems employed in self-driving vehicles (SDVs) aim to understand the scene both spatially and temporally. Recently, 4D panoptic segmentation has emerged as an important task which involves assigning a semantic label to each observation, as well as an instance ID representing each unique object consistently over time, thus combining semantic segmentation, instance segmentation and object tracking into a single, comprehensive task. Potential applications of this task include building semantic maps, auto-labelling object trajectories, and onboard perception. The task is, however, challenging due to the sparsity of the point-cloud observations, and the computational complexity of 4D spatio-temporal reasoning.

Traditionally, researchers have tackled the constituent tasks in isolation, *i.e.*, segmenting classes [1, 2, 3, 4], identifying individual objects [5, 6], and tracking them over time [7, 8]. However, combining multiple networks into a single perception system makes it error-prone, potentially slow, and cumbersome to train. Recently, end-to-end approaches [9, 10, 11] for 4D panoptic segmentation have emerged, but they utilize only LiDAR data which provides accurate 3D geometry, but is sparse at range and lacks visual appearance information that might be important to disambiguate certain classes (e.g., a pedestrian might look like a pole at range). Nonetheless, combining LiDAR and camera data effectively and efficiently is non-trivial as the observations are very different in nature.

In this paper, we propose 4D-Former, a novel approach for 4D panoptic segmentation that effectively fuses information from LiDAR and camera data to output high quality semantic segmentation labels as well as temporally consistent object masks for the input point cloud sequence. To the best of our knowledge, this is the first work that explores multi-sensor fusion for 4D panoptic point cloud segmentation. Towards this goal, we propose a novel transformer-based architecture that

---

∗Indicates equal contribution. †Work done while an intern at Waabi.

fuses features from both modalities by efficiently encoding object instances and semantic classes as concise *queries*. Moreover, we propose a learned tracking framework that maintains a history of previously observed object tracks, allowing us to overcome occlusions without hand-crafted heuristics. This gives us an elegant way to reason in space and time about all the tasks that constitute 4D panoptic segmentation. We demonstrate the effectiveness of 4D-Former on both nuScenes [12] and SemanticKITTI [13] benchmarks and show that we significantly outperform the state-of-the-art.

## 2    Related Work

**3D Panoptic Segmentation:**    This task combines semantic and instance segmentation, but does not require temporally consistent object tracks. Current approaches often utilize a multi-branch architecture to independently predict semantic and instance labels. A backbone network is used to extract features from the LiDAR point cloud with various representations *e.g.* points [14], voxels [1, 3], 2D range views [15, 16], or birds-eye views [17]. Subsequently, the network branches into two paths to generate semantic and instance segmentation predictions. Typically, instance predictions are obtained through deterministic [18, 19, 20] or learnable clustering [6, 21], proposal generation [22], or graph-based methods [23, 24]. These methods are not optimized end-to-end. Several recent work [25, 26, 27] extends the image-level approach from Cheng *et al.* [28] to perform panoptic segmentation in the LiDAR domain in an end-to-end fashion. We adopt a similar approach to predict semantic and instance masks from learned queries, however, our queries attend to multi-modal features whereas the former utilizes only LiDAR inputs.

**LiDAR Tracking:**    This task involves predicting temporally consistent bounding-boxes for the objects in the input LiDAR sequence. We classify existing approaches into two main groups: tracking-by-detection and end-to-end methods. The tracking-by-detection paradigm [7, 8, 29] has been widely researched, and generally consists of a detection framework followed by a tracking mechanism. Since LiDAR point clouds typically lack appearance information but offer more spatial and geometric cues, existing approaches usually rely on motion cues for tracking (*e.g.* Kalman Filters [30], Hungarian matching [31] or Greedy Algorithm [8] for association). Recently, end-to-end frameworks [32] have also emerged where a single network performs per-frame detection and temporal association. In contrast to these, 4D-Former utilizes both LiDAR and image modalities, and performs point-level instance tracking and semantic segmentation with a single unified framework.

**4D Panoptic Segmentation:**    This is the task we tackle in our work, and it involves extending 3D panoptic segmentation to include temporally consistent instance segmentation throughout the input sequence. Most existing methods [9, 11, 33] employ a sliding-window approach which tracks instances within a short clip of upto 5 frames. 4D-PLS [9] models object tracklets as Gaussian distributions and segments them by clustering per-point spatio-temporal embeddings over the 4D input volume. 4D-StOP [11] proposes a center-based voting technique to generate track proposals which are then aggregated using learned geometric features. These methods associate instances across clips using mask IoU in overlapping frames. CA-Net [10], on the other hand, learns contrastive embeddings for objects to associate per-frame predictions over time. Recently, concurrent work from Zhu *et al.* [34] develops rotation-equivariant networks which provide more robust feature learning for 4D panoptic segmentation. Different to these, 4D-Former utilizes multimodal inputs, and adopts a transformer-based architecture which models semantic classes and objects as concise *queries*.

**LiDAR and Camera Fusion:**    Multimodal approaches have recently become popular for object detection and semantic segmentation. Existing methods can be grouped into two categories: (1) point-level fusion methods, which typically involve appending camera features to each LiDAR point [35, 36, 37] or fusing the two modalities at the feature level [38, 39, 40]. (2) Proposal-level fusion, where object detection approaches [41, 42] employ transformer-based architectures which represent object as queries and then fuse them with camera features. Similarly, Li *et al.* [43] perform semantic segmentation by modeling semantic classes as queries which attend to scene features from both modalities. 4D-Former, on the other hand, tackles 4D panoptic segmentation whereas the aforementioned methods perform single-frame semantic segmentation or object detection.

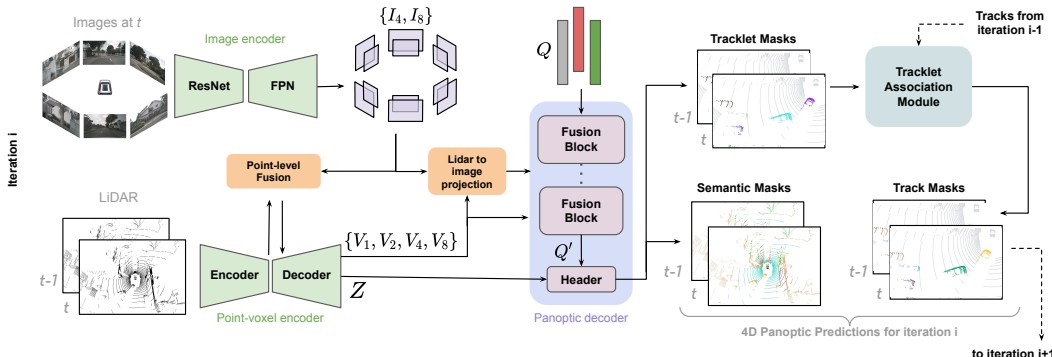

Figure 1: **4D-Former inference** at iteration $i$. Note that tracking history from $i-1$ is used in the Tracklet Association Module.

## 3   Multimodal 4D Panoptic Segmentation

In this paper we propose 4D-Former to tackle 4D panoptic segmentation. The task consists of labelling each 4D LiDAR point with a semantic class and a track ID that specifies a consistent instance over time. Camera images provide rich additional context to help make more accurate predictions, particularly in regions where LiDAR is sparse. To this end, we propose a novel transformer-based architecture that effectively combines sparse geometric features from LiDAR with dense contextual features from cameras. In particular, it models object instances and semantic classes using concise, learnable *queries*, followed by iterative refinement by self-attention and cross-attention to LiDAR and camera image features. Using these queries, our method is able to attend only to regions of the sensor data that are relevant, making the multimodal fusion of multiple cameras and LiDAR tractable. In order to handle sequences of arbitrary length as well as continuous streams of data (e.g., in the onboard setting), 4D-Former operates in a sliding window fashion, as illustrated in Fig. 1. At each iteration, 4D-Former takes as input the current LiDAR scan at time $t$, the past scan at $t-1$, and the camera images at time $t$. It then generates semantic and tracklet predictions for these two LiDAR scans. To make the tracklet predictions consistent over the entire input sequence, we propose a novel Tracklet Association Module (TAM) which maintains a history of previously observed object tracks, and associates them based on a learning-based matching objective.

### 3.1   Multimodal Encoder

Our input encoder extracts image features from the camera images, and point-level and voxel-level features by fusing information from the LiDAR point clouds and camera features. These features are then utilized in our transformer-based panoptic decoder presented in Sec. 3.2.

**Image feature extraction:**   Assume the driving scene is captured by a set of images of size $H \times W$ captured from multiple cameras mounted on the ego-vehicle. We employ a ResNet-50 [44] backbone, followed by a Feature Pyramid Network (FPN) [45], to produce a set of multi-scale, $D-$dimensional feature maps $\{I_s \mid s = 4, 8\}$ for each of the images, where $I_s \in \mathbb{R}^{H/s \times W/s \times D}$.

**Point/voxel feature extraction:**   The network architecture is inspired by [46] and consists of a point-branch and a voxel-branch. The point-branch learns point-level embeddings, thus preserving fine details, whereas the voxel-branch performs contextual reasoning using 3D sparse convolutional blocks [47] and provides multi-scale feature maps. Each of the $N$ points in the input LiDAR

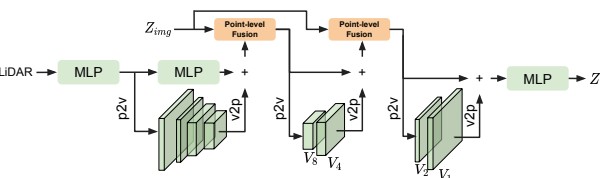

Figure 2: Overview of point and voxel feature extraction. p2v: point-to-voxel. v2p: voxel-to-point.

point-cloud is represented as an 8-D feature which include the $xyz$ coordinates, relative timestamp, intensity, and 3D relative offsets to the nearest voxel center. An MLP is applied to obtain initial point

embeddings which are then averaged over a voxel to obtain voxel features. These voxel features are processed through four residual blocks with 3D sparse convolutions, each of which downsamples the feature map by $2\times$. Four additional residual blocks are then used to upsample the sparse feature maps back to the original resolution, thus yielding a set of $D$-dimensional voxel features at various strides $\mathcal{V} = \{V_i \in \mathbb{R}^{N_i \times D} \mid i = 1, 2, 4, 8\}$, where $N_i$ denotes the number of non-empty voxels at the $i$-th stride. At various stages in this network, point-level features are updated with image features via point-level fusion (as explained in the next paragraph). Moreover, we exploit point-to-voxel and voxel-to-point operations to fuse information between the point and voxel branches at different scales, as illustrated in Fig. 2. We denote the final point-level features as $Z \in \mathbb{R}^{N \times D}$.

**Point-level fusion:** We enrich the geometry-based LiDAR features with appearance-based image features by performing a fine-grained, point-level feature fusion. This is done by taking the point features $Z_{\text{lidar}} \in \mathbb{R}^{N \times D}$ at intermediate stages inside the LiDAR backbone, and projecting their corresponding $(x, y, z)$ coordinates to the highest resolution image feature map $I_4$. Note that this can be done since the image and LiDAR sensors are calibrated, which is typically the case in modern self-driving vehicles. This yields a set of image features $Z_{\text{img}} \in \mathbb{R}^{M \times D}$, where $M \leq N$ since generally not all LiDAR points have valid image projections. We use $Z_{\text{lidar}}^+ \in \mathbb{R}^{M \times D}$ to denote the subset of features in $Z_{\text{lidar}}$ which have valid image projections, and $Z_{\text{lidar}}^- \in \mathbb{R}^{(N-M) \times D}$ for the rest. We then perform point-level fusion between image and LiDAR features as follows:

$$Z_{\text{lidar}}^+ \leftarrow \texttt{MLP}_{\text{fusion}}([Z_{\text{lidar}}^+, Z_{\text{img}}]) \qquad\qquad Z_{\text{lidar}}^- \leftarrow \texttt{MLP}_{\text{pseudo}}(Z_{\text{lidar}}^-) \qquad (1)$$

where both MLPs contain 3 layers, and $[\cdot, \cdot]$ denotes channel-wise concatenation. Intuitively, $\texttt{MLP}_{\text{fusion}}$ performs pairwise fusion for corresponding image and LiDAR features. On the other hand, $\texttt{MLP}_{\text{pseudo}}$ updates the non-projectable LiDAR point features to resemble fused embeddings.

## 3.2 Transformer-based Panoptic Decoder

We propose a novel decoder which predicts per-point semantic and object track masks with a unified architecture. This stands in contrast with existing methods [9, 11, 48, 6] which generally have separate heads for each output. Our architecture is inspired by image-level object detection/segmentation methods [49, 28], but the key difference is that our decoder performs multimodal fusion.

We initialize a set of queries $Q \in \mathbb{R}^{T \times D}$ randomly at the start of training where the number of queries ($T$) is assumed to be an upper-bound on the number of objects in a given scene. The idea to use these queries to segment a varying number of objects as well as the non-instantiable 'stuff' classes in the scene. The queries are input to a series of 'fusion blocks'. Each block is composed of multiple layers where the queries $Q$ are updated by: (1) cross-attending to the voxel features $V_i \in \mathbb{R}^{N_i \times C}$ at a given stride, (2) cross-attending to the set of image

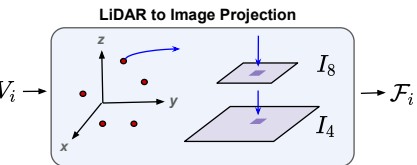

Figure 3: LiDAR to image projection.

features $\mathcal{F}_i \in \mathbb{R}^{M_i \times C}$ which are obtained by projecting the $(x, y, z)$ coordinates for the voxel features $V_i$ into each of the multi-scale image feature maps [1] $\{I_4, I_8\}$ (see Fig. 3 for an illustration), and (3) self-attending to each other twice intermittently, and also passing through $2\times$ Feedforward Networks (FFN). The architecture of these fusion blocks is illustrated in Fig. 4.

These queries distill information about the objects and semantic classes present in the scene. To this end, self-attention enables the queries to exchange information between one another, and cross-attention allows them to learn global context by attending to the features from both modalities across the entire scene. This mitigates the need for dense feature interaction between the two modalities which, if done naively, would be computationally intractable since $N_i$ and $M_i$ are on the order of $10^4$. Our

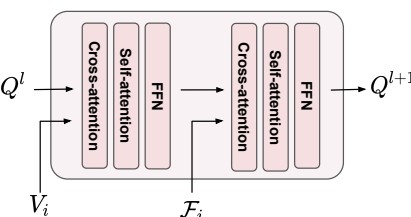

Figure 4: Fusion block architecture.

---

[1] $M_i \leq 2N_i$ since each voxel feature is projected into 2 image feature maps ($I_4$, $I_8$), but not all LiDAR voxel features have valid image projections

fusion block avoids this by leveraging a set of concise queries which attend to the scene features from both modalities in a sequential fashion where the computational footprint of each operation is manageable since $T \ll N_i$ and $T \ll M_i$.

Our transformer-based panoptic decoder is composed of four such fusion blocks, each involving cross-attention to voxel features at different strides, and their corresponding image features. We proceed in a coarse-to-fine manner where the inputs to the fusion blocks are ordered as: $(V_8, \mathcal{F}_8), (V_4, \mathcal{F}_4), (V_2, \mathcal{F}_2), (V_1, \mathcal{F}_1)$. Note that this query-level fusion compliments the fine-grained, point-level fusion in the LiDAR backbone explained in Sec. 3.1. The updated queries output by the decoder, denoted by $Q' \in \mathbb{R}^{T \times D}$, are used to obtain logits for the object tracklet and semantic masks, where each logit represents the log probability of a Bernoulli distribution capturing whether the query represents a specific instance or class. Per-point object tracklet masks $M_p$ are calculated as the dot-product of the updated queries $Q'$ with the point-level features $Z \in \mathbb{R}^{N \times D}$:

$$M_p \leftarrow Z \cdot Q'^T \in \mathbb{R}^{N \times T} \tag{2}$$

Semantic (per-class) confidence scores are obtained by passing $Q'$ through a linear layer. This layer has a fan-out of $1 + C$ to predict a classification score for each of the $C$ semantic classes, and an additional 'no-object' score which is used during inference to detect inactive queries that represent neither an object nor a 'stuff' class. We use the semantic prediction to decide whether the query mask belongs to an object track, or to one of the 'stuff' classes.

**Soft-masked Cross-attention:** Inspired by [50, 28], we employ soft-masked cross-attention to improve convergence. Given a set of queries $Q$, the output $Q_{\text{x-attn}}$ of cross-attention is computed as:

$$Q_{\text{x-attn}} \leftarrow \texttt{softmax}\left(\frac{Q(K + E)^{\text{T}} + \alpha M_v^{\text{T}}}{\sqrt{D}}\right) V \tag{3}$$

Here, $K \in \mathbb{R}^{\{N_i, M_i\} \times D}$ and $V \in \mathbb{R}^{\{N_i, M_i\} \times D}$ denote the keys and values (derived as linear projections from $V_i$ or $\mathcal{F}_i$), respectively, $E \in \mathbb{R}^{\{N_i, M_i\} \times D}$ denotes positional encodings (explained in the next paragraph), $\alpha$ is a scalar weighting factor, and $M_v^T$ is the voxel-level query mask computed by applying Eq. 2 to $Q$, followed by voxelization and downsampling to the required stride. Intuitively, the term "$\alpha M_v^T$" amplifies the correspondence between queries and voxel/image features based on the mask prediction from the previous layer. This makes the queries focus on their respective object/class targets.

**Positional Encodings:** We impart the cross-attention operation with 3D coordinate information of the features in $V_i$ by using positional encodings ($E$ in Eq. 3). These contain two components: (1) Fourier encodings [51] of the $(x, y, z)$ coordinates, and (2) a depth component which is obtained by applying sine and cosine activations at various frequencies to the Euclidean distance of each voxel feature from the LiDAR sensor. Although the depth can theoretically be inferred from the $xyz$ coordinates, we find it beneficial to explicitly encode it. Intuitively, in a multi-modal setup the depth provides a useful cue for how much the model should rely on features from different modalities, *e.g.*, for far-away points the image features are more informative as the LiDAR is very sparse. Both components have $\frac{D}{2}$ dimensions and are concatenated to obtain the final positional encoding $E \in \mathbb{R}^{\{N_i, M_i\} \times D}$. For the image features $\mathcal{F}_i$, we use the encoding of the corresponding voxel.

### 3.3 Tracklet Association Module (TAM)

The 4D panoptic task requires object track IDs to be consistent over time. Since 4D-Former processes overlapping clips, one way to achieve temporal consistency is to associate tracklet masks across clips based on their respective mask IoUs, as done by existing works [9, 11, 33]. However, this approach cannot resolve even brief occlusions which frequently arise due to inaccurate mask predictions and/or objects moving out of view. To mitigate this shortcoming, we propose a learnable Tracklet Association Module (TAM) which can associate tracklets across longer frame gaps and reasons over the objects' appearances and spatial locations.

The TAM is implemented as an MLP which predicts an association score for a given pair of tracklets. The input to our TAM is constructed by concatenating the following attributes of the input tracklet

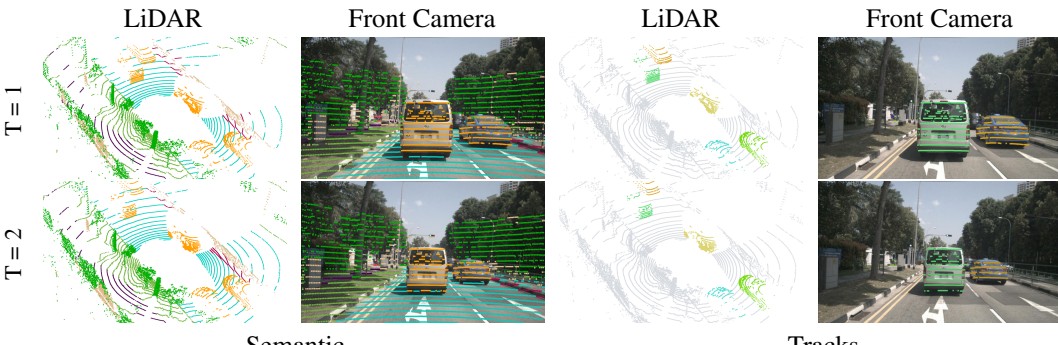

| LiDAR | Front Camera | LiDAR | Front Camera |

Figure 5: Qualitative results on nuScenes sequence 0798 with both LiDAR and image views.

pair along the feature dimension: (1) their $(x, y, z)$ mask centroid coordinates, (2) their respective tracklet queries, (3) the frame gap between them, and (4) their mask IoU. We refer the readers to the supplementary material for an illustration. Intuitively, the tracklet queries encode object appearances, whereas the frame gap, mask centroid and mask IoU provide strong spatial cues. Our TAM contains 4 fully connected layers and produces a scalar association score as the final output. The mask IoU is set to zero for tracklet pairs with no overlapping frames. Furthermore, the mask centroid coordinates and frame gap are expanded to 64-D each by applying sine and cosine activations at various frequencies, similar to the depth encodings discussed in Sec. 3.2.

During inference, we maintain a memory bank containing object tracks. Each track is represented by the tracklet query, mask centroid and frame number for its most recent occurrence. This memory bank is maintained for the past $T_{\text{hist}}$ frames. At frame $t$, we compute association scores for all pairwise combinations of the tracks in the memory bank with the predicted tracklets in the current clip. Subsequently, based on the association score, each tracklet in the current clip is either associated with a previous track ID, or is used to start a new track ID.

### 3.4 Learning

We employ a two-stage training approach for the proposed architecture. During the first stage, we exclude the TAM and optimize the network for a single input clip. The predicted masks are matched to ground-truth objects and stuff class segments with bi-partite matching based on their mask IoUs and classification scores. Note that during training, each stuff class present in the scene is treated as an object track. Subsequently, the masks are supervised with a combination of binary cross entropy and DICE losses, denoted by $\mathcal{L}_{\text{ce}}$ and $\mathcal{L}_{\text{dice}}$ respectively, and the classification output is supervised with a cross-entropy loss $\mathcal{L}_{\text{cls}}$. These losses are computed for the output of each of the $B$ fusion blocks in the Panoptic Decoder, followed by summation. Lastly, the point-level pseudo-fusion output discussed in Sec. 3.1 is supervised by the following $L_2$ regression loss $\mathcal{L}_{\text{pf}}$:

$$\mathcal{L}_{\text{pf}} \leftarrow \left|\left| \text{MLP}_{\text{fusion}}([Z_{\text{lidar}}^+, Z_{\text{img}}]) - \text{MLP}_{\text{pseudo}}(Z_{\text{lidar}}^+) \right|\right|^2 \tag{4}$$

The final loss $\mathcal{L}_{\text{total}}$ is computed by taking the sum of the individual losses with empirically chosen weights. The superscript $\mathcal{L}_{\cdot}^b$ denotes the loss for the output of the $b$-th fusion block in the decoder.

$$\mathcal{L}_{\text{total}} \leftarrow +\mathcal{L}_{\text{pf}} + \sum_{b=1}^{B} \left( 5\,\mathcal{L}_{\text{ce}}^b + 2\,\mathcal{L}_{\text{dice}}^b + 2\,\mathcal{L}_{\text{cls}}^b \right) \tag{5}$$

The second stage involves optimizing the TAM with the remaining network frozen. We generate tracklet predictions for multiple clips separated by different frame gaps, and then optimize the TAM using all pairwise combinations of tracklets in the given clip set. The predicted association scores are supervised with a binary cross-entropy loss.

## 4 Experiments

**Implementation Details:** We process clips containing 2 frames each, with voxel size of 0.1 m, and the feature dimensionality $D = 128$. The images are resized in an aspect-ratio preserving

| Method | Validation | | | | | | Test | | | | | |
|---|---|---|---|---|---|---|---|---|---|---|---|---|
| | PAT | LSTQ | PTQ | PQ | TQ | $S_{assoc}$ | PAT | LSTQ | PTQ | PQ | TQ | $S_{assoc}$ |
| PanopticTrackNet [53] | 44.0 | 43.4 | 50.9 | 51.6 | 38.5 | 32.3 | 45.7 | 44.8 | 51.6 | 51.7 | 40.9 | 36.7 |
| 4D-PLS [9] | 59.2 | 56.1 | 55.5 | 56.3 | 62.3 | 51.4 | 60.5 | 57.8 | 55.6 | 56.6 | 64.9 | 53.6 |
| Cylinder3D++ [1] + OGR3MOT [54] | - | - | - | - | - | - | 62.7 | 61.7 | 61.3 | 61.6 | 63.8 | 59.4 |
| $(AF)^2$-S3Net [3] + OGR3MOT [54] | - | - | - | - | - | - | 62.9 | 62.4 | 60.9 | 61.3 | 64.5 | 59.9 |
| EfficientLPS [22] + KF [30] | 64.6 | 62.0 | 60.6 | 62.0 | 67.6 | 58.6 | 67.1 | 63.7 | 62.3 | 63.6 | 71.2 | 60.2 |
| EfficientLPT [33] | - | - | - | - | - | - | 70.4 | 66.0 | 67.0 | 67.9 | 71.2 | - |
| 4D-Former | **78.3** | **76.4** | **75.2** | **77.3** | **79.4** | **73.9** | **79.4** | **78.2** | **75.5** | **78.0** | **75.5** | **76.1** |

Table 1: Benchmark results for nuScenes validation and test set.

manner such that the lower dimension is 480px. For training time data augmentation, we randomly subsample the LiDAR pointcloud to $10^5$ points, and also apply random rotation and point jitter. The images undergo SSD-based color augmentation [52], and are randomly cropped to 70% of their original size. In the first stage, we train for 80 epochs with AdamW optimizer with batch size 8 across 8x Nvidia T4 GPUs (14GB of usable VRAM). The learning rate is set to $3 \times 10^{-3}$ for the LiDAR feature extractor, and $10^{-4}$ for the rest of the network. The rate is decayed in steps of 0.1 after 30 and 60 epochs. For the second stage, we train the TAM for 2 epochs on a single GPU with learning rate $10^{-4}$. During inference, we associate tracklets over temporal history $T_{hist} = 4$.

**Datasets:** To verify the efficacy of our approach, we apply it to two popular benchmarks: nuScenes [12] and SemanticKITTI [13]. nuScenes [12] contains 1000 sequences, each 20s long and annotated at 2Hz. The scenes are captured with a 32-beam LiDAR sensor and 6 cameras mounted at different angles around the ego vehicle. The training set contains 600 sequences, whereas validation and test each contain 150. The primary evaluation metric is Panoptic Tracking (PAT). Compared to nuScenes, SemanticKITTI [9] contains fewer but longer sequences, and uses LiDAR Segmentation and Tracking Quality (LSTQ) as the primary evaluation metric. One caveat is that image input is only available from a single, forward-facing camera. As a result, only a small fraction ($\sim 15\%$) of LiDAR points are visible in the camera image. For this reason, following existing multimodal methods [43], we evaluate only those points which have valid camera image projections.

**Comparison to state-of-the-art:** Results on nuScenes are shown in Tab. 1 and visualized in Fig. 5. We see that 4D-Former outperforms existing methods across all metrics. In terms PAT, 4D-Former achieves 78.3 and 79.4 on the val and test sets, respectively. This is significantly better than the 70.4 (+9.0) achieved by EfficientLPT [33] on the test set and the 64.6

| Method | LSTQ | $S_{assoc}$ | $S_{cls}$ | $IoU^{st}$ | $IoU^{th}$ |
|---|---|---|---|---|---|
| 4D-PLS [9] | 65.4 | 72.3 | 59.1 | 62.6 | 61.8 |
| 4D-StOP [11] | 71.0 | **82.5** | 61.0 | 63.0 | 66.0 |
| Ours | **73.9** | 80.9 | **67.6** | **64.9** | **71.3** |

Table 2: SemanticKITTI validation results.

(+13.7) achieved by EfficientLPS [22]+KF on val. We attribute this to 4D-Former's ability to reason over multimodal inputs and segment both semantic classes and object tracks in an end-to-end learned fashion. The results on SemanticKITTI validation set are reported in Tab. 2. For a fair comparison, we also evaluated existing top-performing methods on the same sub-set of camera-projectable points. We see that 4D-Former achieves 73.9 LSTQ which is higher than the 71.0 (+2.8) achieved by 4D-StOP and also the 65.4 (+8.4) achieved by 4D-PLS [9]. Aside from $S_{assoc}$, 4D-Former is also better for other metrics.

**Effect of Tracklet Association Module:** The effectiveness of the TAM is evident from Tab. 3 where we compare it to a baseline which uses only use mask IoU in the overlapping frame for association. This results in the PAT dropping from 78.3 to 76.3. This highlights the importance of using a learned temporal association mechanism with both spatial and appearance cues.

| Setting | PAT | LSTQ | PTQ | PQ |
|---|---|---|---|---|
| Mask IoU | 76.3 | 74.6 | 73.9 | **77.3** |
| TAM | **78.3** | **76.4** | **75.2** | **77.3** |

Table 3: Ablation results for temporal association on nuScenes validation set.

Next, we ablate other aspects of our method in Tab. 4. For these experiments, we subsample the training set by using only every fourth frame to save time and resources. The final model is also re-trained with this setting for a fair comparison (row 6).

**Effect of Image Fusion:** Row 1 is a LiDAR-only model which does not utilize image input in any way. This achieves 59.7 PAT which is significantly worse than the final model's 66.1. This shows that using image information yields significant performance improvements. Row 2 utilizes point-level fusion (Sec. 3.1), but does not apply cross-attention to image features in the decoder (Sec. 3.2). This setting achieves 61.8 PAT which is better than the LiDAR-only setting (59.7), but still much worse than the final model (66.1). Row 3 tests the opposite configuration: the decoder includes cross-attention

| # | PF | CAF | DE | PC | PAT | LSTQ | PTQ | PQ |
|---|----|----|----|----|------|------|------|------|
| 1. |   |   | ✓ | ✓ | 59.7 | 64.3 | 60.8 | 63.6 |
| 2. | ✓ |   | ✓ | ✓ | 61.8 | 65.2 | 64.3 | 67.6 |
| 3. |   | ✓ | ✓ | ✓ | 63.2 | 66.1 | 63.1 | 66.3 |
| 4. | ✓ | ✓ |   | ✓ | 64.1 | 66.4 | 65.7 | 69.1 |
| 5. | ✓ | ✓ | ✓ |   | 64.6 | 66.7 | 66.0 | 69.4 |
| 6. | ✓ | ✓ | ✓ | ✓ | **66.1** | **67.4** | **66.2** | **69.9** |

Table 4: Ablation results on nuScenes val set. PF: Point Fusion, CAF: Cross-attention Fusion, DE: Depth encodings, PC: Pseudo-camera feature loss.

to image features, but no point-level fusion is applied. This yields 63.2 PAT which is slightly higher than row 2 (61.8) but worse than the final setting (66.1). We conclude that while both types of fusion are beneficial in a standalone setting, combining them yields a larger improvement.

**Effect of Depth Encodings:** As discussed in Sec. 3.2, our positional encodings contain a depth component which is calculated by applying sine/cosine activations with multiple frequencies to the depth value of each voxel feature. Row 4 omits this component and instead only uses Fourier encodings based on the $xyz$ coordinates. This setting yields 64.1 PAT which is lower than the full model (66.1), thus showing that explicitly encoding depth is beneficial.

**Effect of Pseudo-camera Feature Loss:** Recall from Sec. 3.4 that we supervise pseudo-camera features for point fusion with an $L_2$ regression loss. Row 5 shows that without this loss the PAT reduces from 66.1 to 64.6. Other metrics also reduce, though to a lesser extent.

## 5 Limitations

Our method performs less effectively on SemanticKITTI compared to nuScenes, particularly in crowded scenes with several objects. In addition to lower camera image coverage, this is due to the limited number of moving actors in the SemanticKITTI training set which, on average, contains only 0.63 pedestrians and 0.18 riders per frame. Existing LiDAR-only methods [2, 20, 55] overcome this by using instance cutmix augmentation which involves randomly inserting LiDAR scan cutouts of actors into training scenes. Doing the same in a multimodal setting is, however, non-trivial since it would require the camera images to also be augmented accordingly. Consequently, a promising future direction is to develop more effective augmentation techniques for multimodal training.

Our tracking quality is generally good for vehicles, but is comparatively worse for smaller object classes *e.g.* *bicycle*, *pedestrian* (see class-wise results in the supplementary material), and although the TAM is more effective than mask IoU, the improvement plateaus at $T_{\text{hist}} = 4$ (*i.e.* 2s into the past). Another area for future work thus involves improving the tracking mechanism to handle longer time horizons and challenging object classes.

## 6 Conclusion

We proposed a novel, online approach for 4D panoptic segmentation which leverages both LiDAR scans and RGB images. We employ a transformer-based Panoptic Decoder which segments semantic classes and object tracklets by attending to scenes features from both modalities. Furthermore, our Tracklet Association Module (TAM) accurately associates tracklets over time in a learned fashion by reasoning over spatial and appearance cues. 4D-Former achieves state-of-the-art results on the nuScenes and SemanticKITTI benchmarks, thus demonstrating its efficacy on large-scale, real-world data. We hope our work will spur advancement in SDV perception systems, and encourage other researchers to develop multi-sensor methods for further improvement.

**Acknowledgments**

We thank the anonymous reviewers for the insightful comments and suggestions. We would also like to thank the Waabi team for their valuable support.

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

# Supplementary Materials

## A Tracklet Association Module

We provide an illustration of the proposed Tracklet Association Module (TAM) in Fig.6. The input to our TAM is constructed by concatenating the following attributes of the input tracklet pair along the feature dimension: (1) their $(x, y, z)$ mask centroid coordinates, (2) their respective tracklet queries, (3) the frame gap between them, and (4) their mask IoU. The frame gap and mask centroid coordinates are expanded to 64-D each by applying sine/cosine activations with various frequencies. The concatenated set of features is input to a 4-layer MLP which produces a scalar association score for the input tracklet pair.

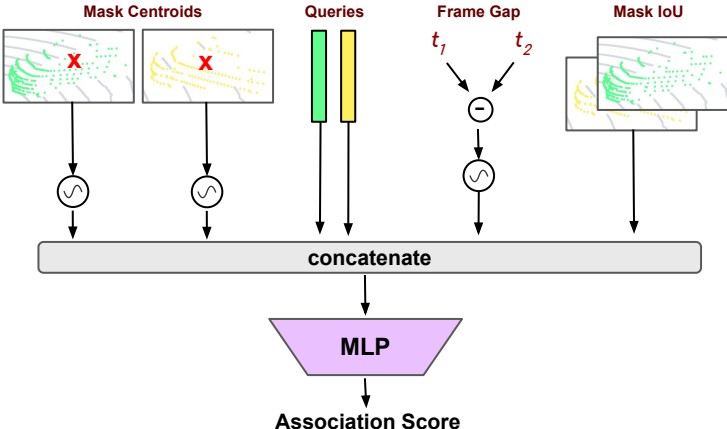

Figure 6: Illustration of the Tracklet Association Module (TAM).

## B Detailed Quantitative Results

In this section, we first present the 3D panoptic metrics on the two benchmarks for reference and then provide the detailed class-wise metrics on the two datasets.

Specifically, we present the 3D panoptic metrics on nuScenes validation set in Tab. 5. Please note that we did not include any other methods in the table since it's unfair to directly compare with other single-scan based methods on the 3D benchmark. For completeness, we evaluate our SemanticKITTI results using 3D panoptic metrics and report the results in Tab. 6 for both cases: evaluating only those points which are projectable into the camera (Camera FoV) and also the Full Scan which includes all LiDAR scan points. Unsurprisingly, because of the missing camera image input, our performance on the full scan (60.7 PQ) is lower than that on the camera FoV only (64.3 PQ). Lastly, we present the detailed per-class results for: nuScenes val set (Tab. 7), nuScenes test set (Tab. 8), and SemanticKITTI val set (Tab.9).

| | PQ | PQ$^{\dagger}$ | PQ$^{St}$ | PQ$^{Th}$ | RQ | RQ$^{St}$ | RQ$^{Th}$ | SQ | SQ$^{St}$ | SQ$^{Th}$ |
|---|---|---|---|---|---|---|---|---|---|---|
| 4D-Former [Ours] | 77.3 | 80.9 | 73.5 | 79.6 | 86.5 | 84.1 | 87.8 | 89.0 | 86.7 | 90.4 |

Table 5: Results on nuScenes 3D panoptic segmentation validation benchmark

| | PQ | PQ$^{\dagger}$ | PQ$^{St}$ | PQ$^{Th}$ | RQ | RQ$^{St}$ | RQ$^{Th}$ | SQ | SQ$^{St}$ | SQ$^{Th}$ | mIoU |
|---|---|---|---|---|---|---|---|---|---|---|---|
| Full Scan | 60.7 | 65.4 | 56.6 | 66.4 | 70.3 | 68.8 | 72.4 | 76.0 | 72.9 | 80.1 | 66.3 |
| Camera FoV only | 64.3 | 66.7 | 60.6 | 69.5 | 73.6 | 72.1 | 75.6 | 80.6 | 80.6 | 80.5 | 67.6 |

Table 6: Results on SemanticKITTI 3D panoptic segmentation validation benchmark

| Metric | mean | Barrier | Bicycle | Bus | Car | Construction | Motorcycle | Pedestrain | Traffic Cone | Trailer | Truck | Drivable | Other Flat | Sidewalk | Terrain | Manmade | Vegetation |
|---|---|---|---|---|---|---|---|---|---|---|---|---|---|---|---|---|---|
| PTQ | 75.17 | 64.11 | 74.33 | 79.05 | 90.89 | 64.64 | 81.87 | 88.03 | 83.04 | 58.67 | 76.92 | 95.61 | 51.92 | 68.92 | 54.61 | 82.46 | 87.59 |
| sPTQ | 75.50 | 65.25 | 75.33 | 79.39 | 91.16 | 64.94 | 82.43 | 88.47 | 83.65 | 59.14 | 77.26 | 95.61 | 51.92 | 68.92 | 54.61 | 82.46 | 87.59 |
| IoU | 78.86 | 82.74 | 52.69 | 90.41 | 94.31 | 54.95 | 88.96 | 82.66 | 68.98 | 65.41 | 82.57 | 96.32 | 71.33 | 73.36 | 75.54 | 91.80 | 89.75 |
| PQ | 77.34 | 68.59 | 79.51 | 80.98 | 93.51 | 67.63 | 86.77 | 91.71 | 87.74 | 61.00 | 78.91 | 95.61 | 51.92 | 68.92 | 54.61 | 82.46 | 87.59 |
| SQ | 89.02 | 82.53 | 87.83 | 93.95 | 95.73 | 88.56 | 91.36 | 93.59 | 90.53 | 86.60 | 93.66 | 96.13 | 84.50 | 79.75 | 78.79 | 91.11 | 89.64 |
| RQ | 86.46 | 83.11 | 90.52 | 86.19 | 97.68 | 76.37 | 94.98 | 98.00 | 96.92 | 70.44 | 84.26 | 99.45 | 61.45 | 86.42 | 69.30 | 90.51 | 97.72 |

Table 7: Class-wise results on nuScenes val set. Metrics are provided in [%]

| Metric | mean | Barrier | Bicycle | Bus | Car | Construction | Motorcycle | Pedestrain | Traffic Cone | Trailer | Truck | Drivable | Other Flat | Sidewalk | Terrain | Manmade | Vegetation |
|---|---|---|---|---|---|---|---|---|---|---|---|---|---|---|---|---|---|
| PTQ | 75.47 | 63.20 | 73.20 | 75.21 | 90.14 | 62.44 | 81.01 | 89.11 | 84.95 | 65.46 | 75.13 | 97.10 | 46.13 | 71.44 | 58.00 | 85.16 | 89.85 |
| sPTQ | 75.90 | 64.63 | 73.98 | 75.42 | 90.45 | 63.73 | 81.92 | 89.57 | 85.48 | 66.14 | 75.43 | 97.10 | 46.13 | 71.44 | 58.00 | 85.16 | 89.85 |
| IoU | 80.42 | 86.66 | 48.99 | 92.24 | 91.72 | 68.22 | 79.79 | 79.84 | 77.24 | 85.54 | 73.81 | 97.41 | 66.51 | 78.50 | 76.62 | 93.04 | 90.62 |
| PQ | 77.99 | 68.63 | 78.30 | 77.48 | 93.01 | 69.07 | 86.69 | 92.64 | 89.13 | 68.17 | 77.05 | 97.10 | 46.13 | 71.44 | 58.00 | 85.16 | 89.85 |
| SQ | 89.66 | 81.69 | 89.13 | 94.74 | 95.80 | 87.12 | 92.62 | 93.94 | 91.63 | 88.30 | 94.29 | 97.36 | 85.46 | 81.85 | 78.04 | 91.08 | 91.51 |
| RQ | 86.59 | 84.01 | 87.85 | 81.78 | 97.09 | 79.28 | 93.60 | 98.62 | 97.28 | 77.19 | 81.71 | 99.73 | 53.98 | 87.28 | 74.31 | 93.50 | 98.19 |

Table 8: Class-wise results on nuScenes test set. Metrics are provided in [%]

# C   Qualitative Comparison (LiDAR-only vs. Fusion)

Figures 7 and 8 provide a qualitative comparison of our proposed method with the LiDAR-only baseline (Tab. 4, row 1 in the main text). We provide the segmentation results in the LiDAR domain for both LiDAR-only and fusion models in the first two columns, respectively, and the corresponding camera view in the third column. The region of interest in each case is highlighted in red.

In the first example (Fig. 7), the baseline wrongly segments the building at range as vegetation due to the limited information obtained from the LiDAR input. By contrast, the final model with fusion effectively leverages the rich contextual information from the camera (highlighted by the red box) and segments the correct class.

In the second example (Fig. 8), the baseline fails to track pedestrians when they are close to each other (the two pedestrians on the left are merged together as a single instance). By contrast, the camera view provides distinct appearance cues for each pedestrian, enabling our model to accurately segment and track them.

| Metric | mean | Car | Bicycle | Motorcycle | Truck | Other Vehicle | Person | Bicyclist | Motorcyclist | Road | Parking | Sidewalk | Other Ground | Building | Fence | Vegetation | Trunk | Terrain | Pole | Traffic Sign |
|---|---|---|---|---|---|---|---|---|---|---|---|---|---|---|---|---|---|---|---|---|
| Assoc | 80.9 | 89.0 | 32.0 | 63.0 | 88.0 | 56.0 | 49.0 | 82.0 | 31.0 | - | - | - | - | - | - | - | - | - | - | - |
| IoU | 67.6 | 97.0 | 61.0 | 78.0 | 84.0 | 73.0 | 83.0 | 95.0 | 0.0 | 96.0 | 44.0 | 80.0 | 4.0 | 88.0 | 56.0 | 89.0 | 71.0 | 76.0 | 66.0 | 45.0 |

Table 9: Class-wise results on SemanticKITTI val set. Metrics are provided in [%]. Note that association metrics are not available for 'stuff' classes.

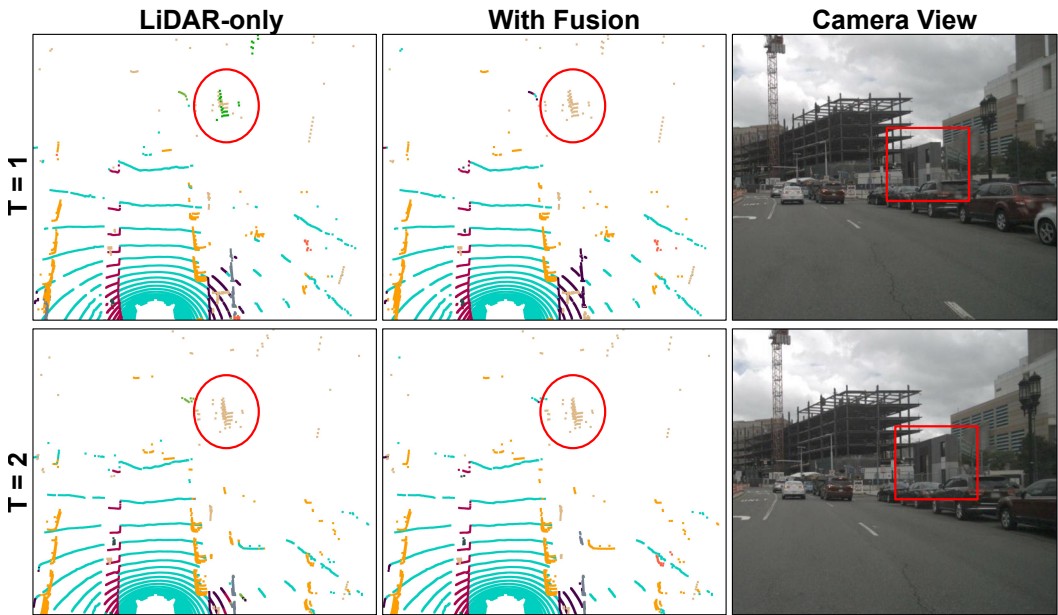

Figure 7: Qualitative comparison of semantic segmentation for LiDAR-only vs. fusion model on sequence 0105 from nuScenes.

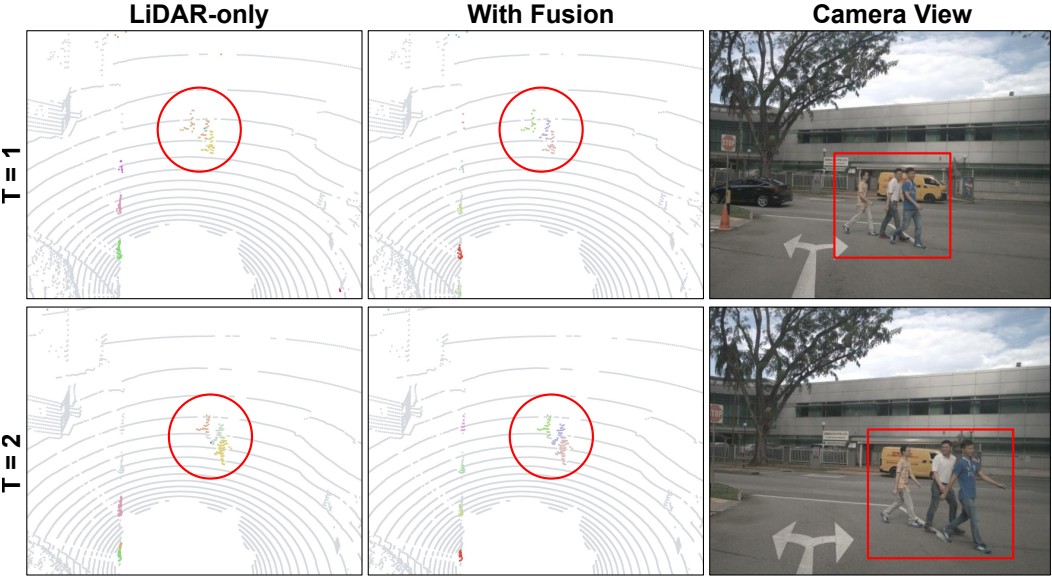

Figure 8: Qualitative comparison of instance segmentation and tracking for LiDAR-only vs. fusion model on sequence 0003 from nuScenes.

