# OpenReview forum: "4D-Former: Multimodal 4D Panoptic Segmentation"
_robot-learning.org/CoRL/2023/Conference — CoRL 2023 Poster_

### Official Review · Reviewer_56js · 2023-07-13

**Confidence:** 5
**Originality:** Good
**Technical Quality:** Good
**Clarity Of Presentation:** Good
**Impact:** 3

**Recommendation:**

Weak Accept: I recommend accepting the paper, but will not argue for my recommendation if the majority of other reviewers have a different opinion.

**Review:**

Strengths:

- This paper proposes the first multi-modality transformer-based framework for 4D LiDAR panoptic segmentation, and it is also effective and practical, as shown in the experiments.
- The basic idea is easy to follow, and the overall methodology is simple yet effective.
- Detailed ablation studies show the importance of each component.
- It achieves new state-of-the-art and surpasses previous methods by a large margin, validating its effectiveness.

Weaknesses:

- There are some missing related works regarding transformer-based methods for LiDAR-based panoptic segmentation, such as PUPS[1] and P3Former[2]. From this perspective, the contribution of a unified query-based decoder for LiDAR-based panoptic segmentation should be weakened. Technically, there are few new things in the multimodal encoder and transformer-based panoptic decoder except for incorporating multimodal data fusion during the procedure.
- (Minor) It would be better to have more ablation studies for alternative model designs. For example, how about sharing several MLP layers in Eq. (1), i.e., using the same MLP layers to encoder point clouds and using additional layers to fuse part of them with image features?
- (Minor) The overall training is two-stage and there are no experiments comparing two-stage and one-stage training, although the author gives the reason and claims end-to-end training may yield better results.

[1] PUPS: Point Cloud Unified Panoptic Segmentation

[2] Position-Guided Point Cloud Panoptic Segmentation Transformer

**Quality Of The Limitations Section:**

Additional details required

**Questions For Rebuttal:**

See the weaknesses.

**Robotics Focus:**

Highly relevant to robotics but no hardware experiments

**Summary Of Paper:**

This paper proposes a multi-modality transformer-based framework for 4D LiDAR-based point clouds panoptic segmentation. Specifically, it takes multi-view images and LiDAR-based point clouds as input, uses a multimodal encoder to fuse image and point features, and subsequently, designs a query-based unified decoder for panoptic segmentation prediction. Furthermore, a tracking module named "Tracklet Association Module (TAM)" is devised to associate tracklets across longer frame gaps in the multi-frame panoptic segmentation problem. The proposed framework achieves new state-of-the-art on two popular benchmarks, nuScenes and SemanticKITTI, and detailed ablation studies show the efficacy of each component.

**Summary Of Recommendation:**

This paper proposes an effective transformer-based framework for multi-modality 4D LiDAR-based panoptic segmentation. Although there are minor issues regarding missing related works and potentially weakened contributions regarding the transformer-based framework, the overall paper has strong performance and detailed ablation studies, making it potentially beneficial to the community. Therefore, I vote for weak accept.

---

> ### Author Response · Authors · 2023-08-10
> **Author Response to Reviewer 56js**
>
> We thank the reviewer for the thoughtful comments on our paper. We are encouraged that they find our framework effective and practical and our methodology easy to follow. Next, we address the comments.
>
> **Q: There are some missing related works regarding transformer-based methods for LiDAR-based panoptic segmentation, such as PUPS[1] and P3Former[2]. From this perspective, the contribution of a unified query-based decoder for LiDAR-based panoptic segmentation should be weakened. Technically, there are few new things in the multimodal encoder and transformer-based panoptic decoder except for incorporating multimodal data fusion during the procedure.**
>
> **A:** We thank the reviewer for pointing out these concurrent works. We were not aware of these works at the time of writing and will include both in our paper for the camera-ready version.
>
> It is worth noting that both are LiDAR-only methods that perform 3D panoptic segmentation. By contrast, 4DFormer is a multimodal architecture that leverages both LiDAR and camera image data, and is able to perform 4D panoptic segmentation i.e. produce temporally consistent object tracks. For this, we use a query-based representation as a means to efficiently reason over both data modalities across the entire scene (see L.167 - L.178), which is something not present in LiDAR-only methods. Nonetheless, we agree with the reviewer that these are definitely relevant works and have added both methods to our manuscript. In terms of benchmark results, 4DFormer achieves 77.3 PQ on the nuScenes validation set which is higher than the 74.7 achieved by PUPS and also the 75.9 achieved by P3Former. This is despite the fact that we optimized our architecture for 4D panoptic metrics (PAT) and not for the single-scan PQ metric.
>
> **Q: (Minor) It would be better to have more ablation studies for alternative model designs. For example, how about sharing several MLP layers in Eq. (1), i.e., using the same MLP layers to encoder point clouds and using additional layers to fuse part of them with image features?**
>
> **A:** Eq. 1 describes the point-level fusion that takes place inside the LiDAR feature extraction network. Here, $MLP_{fusion}$ and $MLP_{pseudo}$ serve different purposes: the former learns to fuse LiDAR and image features, whereas the latter is supervised to match the feature representation produced by $MLP_{fusion}$ as best as possible for LiDAR points which have no image projection (see Eq. 4 for loss supervision and Table 4, row 5 for ablation). It is not possible to share the same MLP for both purposes since the inputs have different dimensionalities: $MLP_{fusion}$ has a fan-in of $D\times 2$ since it receives concatenated LiDAR and image features as input, whereas $MLP_{pseudo}$ has a fan-in of $D$ receives just LiDAR features. We also highlight that these MLPs each contain only 3 layers, and therefore the number of trainable parameters in them (0.3M) is negligible compared to the number of trainable parameters in the overall LiDAR feature extraction network (41M). If the reviewer is still interested, we can train a new model where the last two layers of the MLP are shared and only the first layer is different to accommodate the differing input dimensionalities, however, this will take approximately 1 week to train with our available GPUs.
>
> **Q: (Minor) The overall training is two-stage and there are no experiments comparing two-stage and one-stage training, although the author gives the reason and claims end-to-end training may yield better results.**
>
> **A:** Since we only have GPUs with 14GB VRAM, it is not possible to train the entire model including TAM in a single stage. We have already made extensive use of gradient checkpointing to fit the model for two-stage training. We wrote that end-to-end training may result in improvements to convey that this could be an interesting direction for future work, not with the intention to claim better performance. We will make this more clear in the manuscript.

---

> > ### Comment · Reviewer_56js · 2023-08-15
> > **Response to Rebuttal**
> >
> > I acknowledge that I have read the authors' rebuttal and the other reviews.
> >
> > Thanks for the rebuttal and it addresses some of my concerns. Combined with other reviews, I will keep my original rating and strongly recommend the author add new results to make this paper more solid. In addition, if time and computation resources are available, it would also be interesting to have more discussions about my minor questions in the final version of the paper.

---

> > > ### Author Response · Authors · 2023-08-15
> > > **Author Response to Reviewer 56js**
> > >
> > > Thank you for your feedback. We will include the new results in the final version of the paper.

---

### Official Review · Reviewer_oeTB · 2023-07-14

**Confidence:** 4
**Originality:** Good
**Technical Quality:** Very Good
**Clarity Of Presentation:** Good
**Impact:** 4

**Recommendation:**

Weak Accept: I recommend accepting the paper, but will not argue for my recommendation if the majority of other reviewers have a different opinion.

**Review:**

Originality:

This paper is the first to use monocular images for point cloud 4D panoptic segmentation. From the technical perspective, this paper combined recent successful solutions for 3D point cloud segmentation and multi-view monocular 3D object detection together for this new task, and the authors also proposed a novel method for temporal association of objects. The discussion of related work is proper. A missing reference is the current SOTA of 4D panoptic segmentation on semantic-kitti: EQ-4DSTOP (https://arxiv.org/pdf/2303.15651.pdf). Another work Panoster (https://arxiv.org/abs/2010.15157) presented a post-processing-free instance prediction module, which is also relevant to this paper's contribution. Overall, this paper has decent novelty.

Quality:

The paper presents a well-assembled framework to process multi-modal inputs to conduct the segmentation task. Besides, I found three designs particularly interesting. First, the point-level fusion has a pseudo fusion module for points without RGB correspondence so that the features are better aligned with those with RGB information, with a corresponding loss function. Second, it employed a query-based approach for cross-modal fusion and segmentation. In terms of segmentation, it naturally solved the problem of consistency between instance and semantic labels. Third, the tracklet association module goes beyond IoU-based association, making use of both the semantic and geometric features. The experiments validate the effectiveness of the methodology. Overall, this work is complete in the standard of a conference paper.

Clarity:

The paper is overall well written, but I found the section of soft-masked cross-attention confusing. The K and V do not match the dimension of M_v. Since K and V are of shape TxD, representing the query features (Q also query features), while M_v is the attention matrix between queries and voxels, it is hence unclear whether this module is self-attention or cross-attention, and where this module is placed in the overall network.

Significance:

The results are important. It pushed forward the state-of-the-art 4D panoptic segmentation in nuScenes, and more importantly showed a viable way of handling this task with additional image input.

Relevance:

The panoptic segmentation task with temporal knowledge is a very relevant problem to the robotic community. This paper brings new modality to this task and obtained good results, thus I believe can attract interest of CoRL audience.

Limitations:

A major weakness of this paper is the performance on the SemanticKITTI dataset. The performance is only marginally better than the baselines while having extra monocular image input, which could make people question the effectiveness of the multimodal fusion. However, the authors did provide a paragraph of discussion on the possible reasons, which I found acceptable.

Strengths:

1. First to propose to solve point cloud 4D panoptic segmentation with camera modality.

2. Reasonably designed framework, fusion, prediction, and association modules. Incorporated recent progress in different subfields.

3. Good results on the nuScenes dataset.

Weakness:

1. Confusing introduction of the soft-masked cross attention module.

2. Mediocre SemanticKITTI performance.

**Quality Of The Limitations Section:**

Limitations are addressed clearly

**Questions For Rebuttal:**

1. Please clarify on the soft-masked cross-attention module.

2. Add the missing references.

**Robotics Focus:**

Highly relevant to robotics but no hardware experiments

**Summary Of Paper:**

The task of this paper is to conduct semantic segmentation, instance segmentation and tracking of Lidar point clouds across time, and the paper proposed to incorporate multi-view camera images as input to enhance the features and improve the performance.

**Summary Of Recommendation:**

This paper proposed to solve the 4D panoptic segmentation task in a new way by incorporating extra camera input. The method has good incorporation of existing solutions in building the overall framework, and also proposed novel submodules to improve the performance. The experimental results are good overall, although not as strong on the SemanticKITTI dataset. I would like to see the soft-masked cross attention clarified, and it is a good paper overall.

After rebuttal:

I'd like to keep my score at Weak Accept.

---

> ### Author Response · Authors · 2023-08-10
> **Author Response to Reviewer oeTB**
>
> We thank the reviewer for the detailed comments and suggestions. We are encouraged that they find our design interesting. Next, we address their comments and questions.
>
> **Q:  clarify on the soft-masked cross-attention module**
>
> **A:** The dimensions for $K$, $V$ and $E$ are indeed incorrectly written. We apologize for this mistake and thank the reviewer for pointing this out. The soft-masked cross-attention module allows the queries $Q \in \mathbb{R}^{T\times D}$ to absorb scene information by attending to the LiDAR voxel features $\in \mathbb{R}^{N_i\times D}$ and the corresponding projected camera image features $\in \mathbb{R}^{M_i\times D}$ in a round-robin fashion in the Panoptic Decoder as illustrated in Fig. 4.
>
> The attention operation involves computing the dot-product between the keys and queries. This yields an `attention weight` tensor in $\mathbb{R}^{T\times N_i}$ when attending to LiDAR features or $\mathbb{R}^{T\times M_i}$ when attending to camera image features. In our soft-masked attention, the attention weights are offset by $M_v^T \in [0,1]$ (scaled by $\alpha$), which is the normalized segmentation mask predicted by the previous Fusion Block, and has the same size as the attention weights ($\mathbb{R}^{T\times N_i}$ or $\mathbb{R}^{T\times M_i}$). The underlying idea here is to enable each of the $T$ queries to focus on specific parts of the scene by amplifying the key-query affinity based on the predicted mask from the previous block. Cheng et al. [24] showed that such attention masking accelerates convergence and improves performance for image-level segmentation tasks. We also found that 4D-Former's performance degrades considerably when using vanilla cross-attention (without masking).
>
> To recap, the dimensions for the tensors relevant to soft-masked cross-attention are listed below:
> * The key $(K)$ and value $(V)$ tensors are in $\mathbb{R}^{N_i\times D}$ when attending to LiDAR voxel features, and $\mathbb{R}^{M_i\times D}$ when attending to camera image features.
> * The positional embeddings $E$ have the same size as $K$ and $V$.
> * The queries $Q$ are in $\mathbb{R}^{T\times D}$
> * The masking tensor $M_v^T$ is in $\mathbb{R}^{T\times N_i}$ when attending to LiDAR voxel features or $\mathbb{R}^{T\times M_i}$ when attending to camera image features.
>
> **Q: Mediocre SemanticKITTI performance**
>
> **A:** Regarding SemanticKITTI experiments, this dataset only contains images from a single, front-facing camera as stated in L.269 of the manuscript. This means that only about 15% of the LiDAR points are projectable into the camera image, thus making the dataset not ideal for evaluating a multimodal method such as ours. Nonetheless, we applied our method to SemanticKITTI to show that it can generalize to other datasets and sensor setups. L324-331 in the manuscript also provides our additional analysis on why 4D-Former performs less effectively on SemanticKITTI compared to nuScenes. To support these claims, we compare our results in the `Full Scan` setting (including all LiDAR scan points) against `Camera FoV only` (including only points that are projectable to camera).
>
> |  | LSTQ | S_assoc | S_cls | IoU_stuff | IoU_things |
> | --- | --- | --- | --- | --- | --- |
> | Full Scan | 69.8 | 73.6 | 66.3 | 64.6 | 68.7 |
> | Camera FoV only | 73.9 | 80.9 | 67.6 | 64.9 | 71.3 |
>
> **Q: Add the missing references.**
>
> **A:** We will add EQ-4DSTOP and Panoster in the related works section for camera-ready.

---

> > ### Comment · Reviewer_oeTB · 2023-08-15
> > **Reply to author's response and one more suggestion.**
> >
> > The explanation helped me better understand the soft-masked cross-attention module. The authors should definitely consider revising the paper accordingly to better introduce this module. Besides, it would be nice to have an ablation study on the soft-masked cross attention vs. one that is not using the soft mask offset. I don't think it is mandatory since this design comes from literature but would be nice to have and let readers better understand the effect of your design choices.
> >
> > Another suggestion is that since you use point-level fusion as one module, you may as well name the fusion blocks in the panoptic decoder as query-level fusion, proposal-level fusion, or object-level-fusion in the paper and in Figure 1. A more consistent naming could improve readers' understanding.
> >
> > I think your response regarding SemanticKITTI's performance is reasonable. I do not see the performance as a deal breaker for this paper to be valuable for the readers.

---

> > > ### Author Response · Authors · 2023-08-15
> > > **Author Response to Reviewer oeTB**
> > >
> > > Thank you for your feedback. We will include your suggestions into the camera-ready version of the paper.

---

### Official Review · Reviewer_g5r2 · 2023-07-19

**Confidence:** 5
**Originality:** Fair
**Technical Quality:** Good
**Clarity Of Presentation:** Good
**Impact:** 3

**Recommendation:**

Weak Reject: I recommend rejecting the paper, but will not argue for my recommendation if the majority of other reviewers have a different opinion.

**Review:**

Strengths:
1. Comprehensive Approach: The paper integrates different modules to address the challenges of multimodal 4D panoptic segmentation, making the system complete and sensible.
2. Clear Methodology: The methodology is presented in a straightforward manner, allowing readers to easily follow the proposed approach. Figure 1 provides a clear overview of the system's structure, while subsequent subfigures and sections delve into the details.
3. Analysis of Limitations: The paper acknowledges the limitations of their methods, providing an insightful analysis of the deficiencies and inspiring future work in the field.

Weaknesses:
1. Fragmented Components: The proposed components sometimes lack sufficient insight into the problems they aim to solve. This issue is also evident in the related works section. Instead of highlighting the novelty of specific fields, the paper introduces components from other fields to showcase superiority without thoroughly addressing the underlying problem.
2. Incomplete Experiments: This paper misses some important comparisons and metrics in the experiment part. While the focus is on 4D panoptic segmentation, it is crucial to include comparisons with 3D panoptic segmentation methods, as it serves as a fundamental task for 4D panoptic segmentation. And the metric Panoptic Quality (PQ) is both important in 3D and 4D panoptic segmentation. Besides, 3D panoptic segmentation is more mature. This method, incorporating multi-modal and temporal information, is expected to surpass previous 3D panoptic segmentation to validate its effectiveness. The comparison on the Semantickitti dataset is also missing in terms of PQ.
3. Lack of Related Works: The paper should discuss more related works, particularly in the field of 3D unified panoptic segmentation, which forms an important foundation for this work. Including works such as PUPS (Zhao et al., 2023) and P3Former (Xiao et al., 2023) would enhance the comparison and provide a broader context for the proposed system.


**Quality Of The Limitations Section:**

Limitations are addressed clearly

**Questions For Rebuttal:**

The authors shall address the weaknesses in the Review section.

**Robotics Focus:**

Highly relevant to robotics but no hardware experiments

**Summary Of Paper:**

This paper proposes a system that incorporates 3D panoptic segmentation, 4D tacking, and multi-modal fusion to fulfill multimodal 4D Panoptic Segmentation. It fuses image information in the 3D feature extraction stage and decode stage. It also designs a Tracklet Association Module to associate object tracks.

**Summary Of Recommendation:**

Considering the limitations as stated in the weaknesses, my rating is Weak Reject.

---

> ### Author Response · Authors · 2023-08-10
> **Author Response to Reviewer g5r2**
>
> We thank the reviewer for the insightful comments and valuable feedback of our paper. We are encouraged that the reviewer finds our approach comprehensive, the presentation straightforward and our analysis of limitations insightful and inspiring future work. Next, we address the questions.
>
> **Q: Fragmented Components: The proposed components sometimes lack sufficient insight into the problems they aim to solve. This issue is also evident in the related works section. Instead of highlighting the novelty of specific fields, the paper introduces components from other fields to showcase superiority without thoroughly addressing the underlying problem.**
>
> **A:** We agree with the reviewer that providing insights is important. In our writing, we provided intuitions and insights of the proposed method, as well as the related works. We provided a few examples next. L73-80 highlighted the novelty of the related works proposed for our task (4D panoptic segmentation). L39-44 explained the high-level intuitions of our design for tackling this task. L212-218 explained the intuition of the design behind our tracking association module (TAM). We would like to ask the reviewer to elaborate on this point further since we are not sure precisely what the concern is. We would be happy to add more insights for camera ready, but it would be helpful if the reviewer could point us out to some examples where there is room for improvement so we can improve it.
>
> **Q: Incomplete Experiments: This paper misses some important comparisons and metrics in the experiment part. While the focus is on 4D panoptic segmentation, it is crucial to include comparisons with 3D panoptic segmentation methods, as it serves as a fundamental task for 4D panoptic segmentation. And the metric Panoptic Quality (PQ) is both important in 3D and 4D panoptic segmentation. Besides, 3D panoptic segmentation is more mature. This method, incorporating multi-modal and temporal information, is expected to surpass previous 3D panoptic segmentation to validate its effectiveness. The comparison on the Semantickitti dataset is also missing in terms of PQ.**
>
> **A:** 3D and 4D panoptic segmentation are different tasks with quite different evaluation metrics. 3D panoptic segmentation uses PQ (Panoptic Quality) metric whereas the 4D task uses PAT and LSTQ metrics. The 4D metrics give more weight to temporal association accuracy for object predictions. Since the goal of our paper was to improve 4D panoptic segmentation, we used the most relevant baselines and metrics for this task.
> However, we understand the reviewer’s curiosity and provide 3D panoptic metrics in nuScenes val set for our approach as well as PUPS and P3Former in the table below.
>
> |  | PQ | PQ_dagger | PQ_stuff | PQ_things | RQ | RQ_stuff | RQ_things | SQ | SQ_stuff | SQ_things |
> | --- | --- | --- | --- | --- | --- | --- | --- | --- | --- | --- |
> | PUPS | 74.7 | 77.3 | 73.6 | 75.4 | 83.3 | 85.6 | 81.9 | 89.4 | 85.3 | 91.8 |
> | P3Former | 75.9 | 78.9 | 75.4 | 76.9 | 84.7 | 87.1 | 83.3 | 89.7 | 86.0 | 92.0 |
> | Ours | 77.3 | 80.9 | 73.5 | 79.6 | 86.5 | 84.1 | 87.8 | 89.0 | 86.7 | 90.4 |
>
> We see that our method achieves 77.3 PQ, which is higher than both PUPS (74.7) and P3Former (75.9).
> Regarding SemanticKITTI experiments, this dataset only contains images from a single, front-facing camera as stated in L.269 of the manuscript. This means that only about 15% of the LiDAR points are projectable into the camera image, thus making the dataset not ideal for evaluating a multimodal method such as ours. Nonetheless, we applied our method to SemanticKITTI to show that it can generalize to other datasets and sensor setups. For this we followed Li \textit{et al.} and evaluated only the subset of LiDAR points which are projectable into the image and reported the results in Table 2 of the manuscript.
> For completeness, we evaluate our SemanticKITTI results using 3D panoptic metrics and report the results below for both cases: evaluating only those points which are projectable into the camera (`Camera FoV`) and also the `Full Scan` which includes all LiDAR scan points.
>
> |  | PQ | PQ_dagger | PQ_stuff | PQ_things | RQ | RQ_stuff | RQ_things | SQ | SQ_stuff | SQ_things | mIoU |
> | --- | --- | --- | --- | --- | --- | --- | --- | --- | --- | --- | --- |
> | Full Scan | 60.7 | 65.4 | 56.6 | 66.4 | 70.3 | 68.8 | 72.4 | 76.0 | 72.9 | 80.1 | 66.3 |
> | Camera FoV | 64.3 | 66.7 | 60.6 | 69.5 | 73.6 | 72.1 | 75.6 | 80.6 | 80.6 | 80.5 | 67.6 |
>
> Unsurprisingly, because of the missing camera image input, our performance on the full scan (60.7 PQ) is lower than that on the camera FoV only (64.3 PQ). Since SoTA methods like PUPS do not provide open-source code and/or results, it is unfortunately not possible to compare our results on the `Camera FoV` setting.

---

> ### Author Response · Authors · 2023-08-10
> **Author Response to Reviewer g5r2 Cont'**
>
> **Q: Lack of Related Works: The paper should discuss more related works, particularly in the field of 3D unified panoptic segmentation, which forms an important foundation for this work. Including works such as PUPS (Zhao et al., 2023) and P3Former (Xiao et al., 2023) would enhance the comparison and provide a broader context for the proposed system.**
>
> **A:** We thank the reviewer for the additional references, we will add these concurrent works in our paper.

---

### Official Review · Reviewer_WZbn · 2023-07-20

**Confidence:** 5
**Originality:** Good
**Technical Quality:** Very Good
**Clarity Of Presentation:** Very Good
**Impact:** 3

**Recommendation:**

Strong Accept: I recommend accepting the paper and will argue for my recommendation even if other reviewers hold a different opinion.

**Review:**

- The paper is easy to understand and logically organized into sections.
- The figures complement the text and help in understanding the tasks easily.
-This work is one of the first to address the task of 4D panoptic segmentation in a multi-modal setting.
-4D-Former achieves SOTA results on both Semantic KITTI and Panoptic-nuScenes.
-The ablation experiments exhibit the usefulness of each architecture design choice quantitatively.


**Quality Of The Limitations Section:**

Limitations are addressed clearly

**Questions For Rebuttal:**

The supplementary addresses the question I would have such as qualitative results and details of TAM. Hence, I have no questions.

**Robotics Focus:**

Relevant but unlikely to deploy to hardware in near future

**Summary Of Paper:**

This paper addresses the challenging task 4D panoptic segmentation by utilizing both LiDAR and RGB inputs. To this end, the authors propose a novel method called 4D-Former that leverages both modalities to predict semantic masks along with temporally consistent object masks for the given input sequence. This paper employs concise queries to encode semantic classes and objects from both modalities. Further, this work proposes a learnable association mechanism to associate objects over time that considers both appearance and spatial location.
Lastly, 4D former achieves SOTA results for nuScenes and SemanticKITTI dataset


**Summary Of Recommendation:**

To the best of my knowledge, this study represents the first effort in tackling the multi-modal 4D panoptic segmentation task. The results achieved by the 4D Former model are convincing. It is my anticipation that this work will lay the groundwork for subsequent advancements in the realm of multi-modal research, serving as a good starting point.

---

> ### Author Response · Authors · 2023-08-10
> **Author Response to Reviewer WZbn**
>
> We thank the reviewer for seeing the value of our work. We are encouraged that the reviewer recognizes our paper’s clear organization, pioneering approach to multimodal 4D panoptic segmentation, convincing state-of-the-art results, and anticipates its impact on advancing multimodal research for 4D panoptic segmentation. This recognition encourages us to continue our efforts in advancing this field and delivering impactful research.

---

### Decision · Program_Chairs · 2023-08-30

**Decision:**

Accept (Poster)

**Comment:**

This paper proposes an approach for 4D panoptic segmentation from camera images and LiDAR scans. The reviewers find the paper to be clearly written and addressing an important problem. Some concerns are convincingly addressed in the rebuttal whereas some regarding the incremental nature of the contributions still remain. Nevertheless, all the reviewers agree that the work is technically correct and well executed.